# Retinal Vascular Occlusion after COVID-19 Vaccination: More Coincidence than Causal Relationship? Data from a Retrospective Multicentre Study

**DOI:** 10.3390/jcm11175101

**Published:** 2022-08-30

**Authors:** Nicolas Feltgen, Thomas Ach, Focke Ziemssen, Carolin Sophie Quante, Oliver Gross, Alaa Din Abdin, Sabine Aisenbrey, Martin C. Bartram, Marcus Blum, Claudia Brockmann, Stefan Dithmar, Wilko Friedrichs, Rainer Guthoff, Lars-Olof Hattenbach, Klaus R. Herrlinger, Susanne Kaskel-Paul, Ramin Khoramnia, Julian E. Klaas, Tim U. Krohne, Albrecht Lommatzsch, Sabine Lueken, Mathias Maier, Lina Nassri, Thien A. Nguyen-Dang, Viola Radeck, Saskia Rau, Johann Roider, Dirk Sandner, Laura Schmalenberger, Irene Schmidtmann, Florian Schubert, Helena Siegel, Martin S. Spitzer, Andreas Stahl, Julia V. Stingl, Felix Treumer, Arne Viestenz, Joachim Wachtlin, Armin Wolf, Julian Zimmermann, Marc Schargus, Alexander K. Schuster

**Affiliations:** 1Clinic of Ophthalmology, University Medical Centre Goettingen, 37075 Goettingen, Germany; 2Department of Ophthalmology, University Hospital Bonn, 53127 Bonn, Germany; 3Department of Ophthalmology, University Leipzig, 04103 Leipzig, Germany; 4Centre for Ophthalmology, University Eye Hospital Tübingen, 72076 Tuebingen, Germany; 5Medical School, University Medical Centre Goettingen, 37075 Goettingen, Germany; 6Clinic of Nephrology and Rheumatology, University Medical Centre Goettingen, 37075 Goettingen, Germany; 7Department of Ophthalmology, Saarland University Medical Centre UKS, 66421 Homburg/Saar, Germany; 8Department of Ophthalmology, Vivantes Health Network Ltd., Neukoelln Hospital, 12351 Berlin, Germany; 9Department of Ophthalmology, Hannover Medical School, 30625 Hannover, Germany; 10Department of Ophthalmology, Helios Hospital Erfurt, 99089 Erfurt, Germany; 11Department of Ophthalmology, Universitätsmedizin Rostock, 18057 Rostock, Germany; 12Department of Ophthalmology, Helios HSK Wiesbaden, 65191 Wiesbaden, Germany; 13Charlottenklinik Ophthalmology, 70176 Stuttgart, Germany; 14Department of Ophthalmology, Faculty of Medicine, University Hospital Duesseldorf, 40225 Dusseldorf, Germany; 15Department of Ophthalmology, Ludwigshafen Hospital, 67063 Ludwigshafen am Rhein, Germany; 16Department of Internal Medicine I, Asklepios Klinik Nord, 22417 Hamburg, Germany; 17Department of Ophthalmology, Maerkische Kliniken GmbH, 58515 Luedenscheid, Germany; 18The David J Apple Centre for Vision Research, Department of Ophthalmology, University Hospital Heidelberg, 69120 Heidelberg, Germany; 19Department of Ophthalmology, University Hospital, LMU Munich, 80336 Munich, Germany; 20Department of Ophthalmology, Faculty of Medicine and University Hospital of Cologne, University of Cologne, 50937 Cologne, Germany; 21Department of Ophthalmology, St. Franziskus-Hospital, 48145 Muenster, Germany; 22Department of Ophthalmology, University of Luebeck, 23538 Luebeck, Germany; 23Ophthalmology Department, Hospital Rechts der Isar, Technical University of Munich (TUM), 81675 Munich, Germany; 24Department of Ophthalmology, University Hospital RWTH Aachen, 52074 Aachen, Germany; 25Praxis Prof. Laube, 40212 Duesseldorf, Germany; 26Department of Ophthalmology, University Hospital of Regensburg, 93049 Regensburg, Germany; 27Department of Ophthalmology, Charité-University Medicine Berlin, Corporate Member of Freie Universitaet Berlin, Humboldt-Universitaet zu Berlin and Berlin Institute of Health, 13353 Berlin, Germany; 28Clinic of Ophthalmology, University Hospital Schleswig-Holstein, Kiel Campus, 24105 Kiel, Germany; 29Department of Ophthalmology, University Hospital Carl Gustav Carus, Technical University of Dresden, 01307 Dresden, Germany; 30Municipal Clinic Braunschweig, 38126 Braunschweig, Germany; 31Institute of Medical Biostatistics, Epidemiology and Informatics, University Medical Centre of the Johannes Gutenberg-University Mainz, 55131 Mainz, Germany; 32Department of Ophthalmology, Phillips University of Marburg, 35043 Marburg, Germany; 33Eye Centre, Faculty of Medicine, University of Freiburg, 79106 Freiburg, Germany; 34Department of Ophthalmology, University Medical Centre Hamburg-Eppendorf, 20251 Hamburg, Germany; 35Department of Ophthalmology, University Medical Centre Greifswald, 17475 Greifswald, Germany; 36Department of Ophthalmology, Medical Centre of the Johannes Gutenberg-University Mainz, 55131 Mainz, Germany; 37Department of Ophthalmology, Klinikum Kassel, 34125 Kassel, Germany; 38Department of Ophthalmology, Martin-Luther University Halle-Wittenberg, 06120 Halle (Saale), Germany; 39Department of Ophthalmology, Sankt-Gertrauden Krankenhaus, 10713 Berlin, Germany; 40MHB Medizinische Hochschule Brandenburg, 16816 Neuruppin, Germany; 41Department of Ophthalmology, Ulm University Medical Centre, 89075 Ulm, Germany; 42Department of Ophthalmology, University of Muenster Medical Centre, 48149 Muenster, Germany; 43Asklepios Augenklinik Nord, 22417 Hamburg, Germany; 44Department of Ophthalmology, University Hospital Duesseldorf, Heinrich Heine University, 40225 Duesseldorf, Germany

**Keywords:** retinal vein occlusion, retinal artery occlusion, anterior ischaemic optic neuropathy, infection, vaccination, SARS-CoV-2, COVID-19

## Abstract

Background: To investigate whether vaccination against SARS-CoV-2 is associated with the onset of retinal vascular occlusive disease (RVOD). Methods: In this multicentre study, data from patients with central and branch retinal vein occlusion (CRVO and BRVO), central and branch retinal artery occlusion (CRAO and BRAO), and anterior ischaemic optic neuropathy (AION) were retrospectively collected during a 2-month index period (1 June–31 July 2021) according to a defined protocol. The relation to any previous vaccination was documented for the consecutive case series. Numbers of RVOD and COVID-19 vaccination were investigated in a case-by-case analysis. A case–control study using age- and sex-matched controls from the general population (study participants from the Gutenberg Health Study) and an adjusted conditional logistic regression analysis was conducted. Results: Four hundred and twenty-one subjects presenting during the index period (61 days) were enrolled: one hundred and twenty-one patients with CRVO, seventy-five with BRVO, fifty-six with CRAO, sixty-five with BRAO, and one hundred and four with AION. Three hundred and thirty-two (78.9%) patients had been vaccinated before the onset of RVOD. The vaccines given were BNT162b2/BioNTech/Pfizer (*n* = 221), followed by ChadOx1/AstraZeneca (*n* = 57), mRNA-1273/Moderna (*n* = 21), and Ad26.COV2.S/Johnson & Johnson (*n* = 11; unknown *n* = 22). Our case–control analysis integrating population-based data from the GHS yielded no evidence of an increased risk after COVID-19 vaccination (OR = 0.93; 95% CI: 0.60–1.45, *p* = 0.75) in connection with a vaccination within a 4-week window. Conclusions: To date, there has been no evidence of any association between SARS-CoV-2 vaccination and a higher RVOD risk.

## 1. Introduction

Any correlation between a vaccination against SARS-CoV-2 and retinal vascular occlusion has not yet been adequately investigated. For possible vaccination-related side effects, recent reviews have reported retinal vascular occlusive disease (RVOD) as single cases and case reports, but the incidence is currently unknown.

A survey among neurologists reported a higher number of cerebral sinus and venous thromboses after SARS-CoV-2 vaccination [1], while ophthalmological case reports and small case series described numerous complications in the anterior and posterior eye segments [2,3,4]. The retinal findings cover uveitis, central serous chorioretinopathy, acute retinal necrosis, and retinal vascular alterations. The underlying pathological mechanisms of vascular occlusive diseases are not yet fully understood, but complement-activated thrombotic microangiopathy, hypercoagulable state, and endotheliitis are possible candidates [5,6,7,8,9,10,11].

Due to the urgent need for appropriate vaccines against COVID-19 disease, the approval processes for appropriate vaccines have certainly differed from those of other drug approvals. Side effects after vaccination may, therefore, have not been comprehensively documented and understood, and administering drugs outside of clinical studies carries the risk of over- or underestimating the incidences of serious adverse events. This uncertainty makes it difficult for physicians to adequately inform patients about potential vaccination-related side effects [12,13].

It is, however, still questionable whether an RVOD occurrence during or after COVID-19 vaccination is coincidental or not. 

To further address possible COVID-19 vaccination-related RVODs, we conducted a survey in retina centres across Germany in the summer of 2021 when vaccination rates were at high levels. The aim of our study was to determine the RVOD incidences in patients with and without COVID-19 vaccination and compare them to data from a population-based cohort study.

## 2. Material and Methods

The German Retina Society (www.retinologie.org; accessed on 4 April 2022) invited 50 retina clinics in Germany (for the complete list of centres, see Appendix A) to contribute to a retrospective study on patients presenting with RVOD.

The study period included 1 June 2021–31 July 2021 (2 months), in which SARS-CoV-2 vaccination rates (first or second vaccination) were at high levels in Germany [14]. No additional vaccinations (boosters) were administered in Germany during this time period. 

Included were all patients with (I) newly diagnosed RVOD (subgroups: central retinal vein occlusion (CRVO), branch retinal vein occlusion (BRVO), central retinal artery occlusion (CRAO), branch retinal artery occlusion (BRAO), and anterior ischaemic optic neuropathy (AION)) and (II) the available data on the COVID-19 vaccinations. The data on these patients were collected and entered in a study form. The anonymized information included: type of RVOD (CRVO, BRVO, CRAO, BRAO, or AION); patient age; gender; involved eye; time point of first symptoms and duration of visual complaints; best-corrected visual acuity (BCVA) using the logMAR scale; pre-existing SARS-CoV-2 infection (yes or no); COVID-19 vaccination (yes or no); number of vaccinations (first or second); time point of the onset of symptoms after first and second vaccination (less than 2 weeks, 2–4 weeks, 4–6 weeks, or more than 6 weeks); type of vaccine administered (BNT162b2 (BioNTech/Pfizer), ChadOx1 (AstraZeneca), mRNA-1273 (Moderna), and/or Ad26.COV2.S (Johnson & Johnson)); pre-existing vascular risk factors (arterial hypertension, diabetes mellitus, obesity, smoking, carotid artery stenosis, atrial fibrillation, coagulation disorders, and glaucoma); and the use of anticoagulation medication (vitamin K-dependent drugs, direct oral anticoagulants, and acetylsalicylic acid). 

An association between RVOD frequency and COVID-19 vaccination was considered possible if the event occurred within 4 weeks after COVID-19 vaccination [1].

The data from this study were compared to age- and sex-matched data from the population-based Gutenberg Health Study (GHS). The GHS is an ongoing population-based cohort study that began at the University Medical Centre of Johannes Gutenberg-University Mainz, Germany, in 2007 [15]. Participants were randomly selected from residents’ registration offices (City of Mainz and District of Mainz-Bingen, Germany) stratified by gender, decade of age, and residence (rural vs. urban) and had a baseline age of 35–74 years. The overall participation rate was 55.5% at the initial examination. The GHS was approved by the Medical Ethics Committee of the State Chamber of Medicine of Rhineland Palatinate in Mainz, Germany. All participants gave their written informed consent prior to study inclusion. As part of the 10-year follow-up examination, their COVID-19 vaccination status was surveyed, and date and type of vaccination was recorded.

All research procedures adhered to the tenets of the Declaration of Helsinki, and the study was approved by the Ethics Committee of the Medical Faculty at UMG University of Goettingen, Germany. Patients’ informed consent for study participation was waived, since the data evaluated were anonymous, and the corresponding information on the vaccination status was retrospectively assigned. This study is reported to be in line with the STROBE statement for observational studies [16].

### Statistics

This project took a mixed-methods approach to ensure the highest probability of detecting any association between RVOD and COVID-19 vaccination. 

Case-by-case analysis (descriptive case-only study): Herein, we descriptively investigated the time-dependent accumulation of COVID-19 vaccinations prior to RVOD disease—more specifically, whether there were more patients who received a COVID-19 vaccination shortly before the RVOD disease.Case–control study: In this analysis, we compared the odds of being vaccinated in the last four weeks among patients with RVOD (cases) to controls from the general population recruited by the Gutenberg Health Study (GHS) (age ±5 years and sex-matched). The recruitment of the controls took place between August 2021 and November 2021 (N = 939). For each control, the vaccination status within the 4 weeks prior to the date of the RVOD diagnosis of the corresponding case was analysed and believed not to be affected by the shift in recruitment time. A conditional logistic regression analysis was computed in (I) an unadjusted way and (II) adjusted for obesity (BMI ≥ 30), diabetes, arterial hypertension, smoking, and use of anticoagulation. All RVOD cases were analysed, as were the different entities of retinal vascular occlusions separately. A sensitivity analysis with cases presenting <2 weeks after symptoms onset was carried out.

All data generated or analysed during this study were included in this published article and its Appendix A. All statistical analyses were conducted with R (R version 4.0.0 (24 April 2020), R Core Team (2020). R: A language and environment for statistical computing. R Foundation for Statistical Computing, Vienna, Austria, https://www.R-project.org/; accessed on 2 February 2022).

## 3. Results

The German Retinological Society invited its members in 51 private practices and eye clinics across Germany to participate in this study. Of these, 37 (73%) responded. 

### 3.1. Case-by-Case Analysis

A total of 508 patient files were transmitted, representing a consecutive case series of 37 eye clinics. Four hundred and twenty-one subjects were included where relevant data on age, sex, vascular occlusion type, and their COVID-19 vaccination status (yes/no) were available, as shown in Figure 1. Comparing excluded and included patients, we detected no significant difference in age, gender, time of occlusion, systemic risk factors, and vaccination status. The study participants’ characteristics are illustrated in Table 1. 

Three hundred and twenty-one study participants (76.2%) were vaccinated at least once before the RVOD onset; therefore, in one hundred study participants, only one vaccination was documented. Most patients received BNT162b (BioNTech/Pfizer) (*n* = 221), followed by ChadOx1 (AstraZeneca) (*n* = 57), mRNA-1273 (Moderna) (*n* = 21), and Ad26.COV2.S (Johnson & Johnson) (*n* = 11; unknown vaccine *n* = 11; not vaccinated *n* = 89). Seventy patients (21.8% of the vaccinated patients) were vaccinated within 2 weeks of RVOD onset, 85 (26.5%) 2–4 weeks, 44 (13.7%) between 4 and 6 weeks, and 122 (38.0%) more than 6 weeks before RVOD onset (Table 1, Appendix A and Figure 2). 

Three hundred and six patients (76.7%) visited their ophthalmologist within 2 weeks after first symptoms occurred, three hundred and fifty patients (83.0%) within 4 weeks. Glaucoma was present in 10.7% (range 7.1–14.6%) for the different types of RVOD, as described above. Of all RVOD patients, CRAO patients had the worst-affected visual acuity (Table 1).

We observed a substantial cardiovascular risk profile in our cohort. Arterial hypertension was present in 64.5% (range 54.7–85.5%), diabetes in 18.0% (10.8–30%), obesity in 14.3% (8.5–24.2%), smoking (currently or previously) in 12.3% (6.8–23.2%), carotid artery stenosis in 18.4% (6.9–29.6%), and atrial fibrillation in 11.7% (8.2–14.8%). Among these, CRAO patients were most likely to present cardiovascular risk factors. One hundred and sixty-seven patients (39.7%; range 23.5–61.1%) were on anticoagulation drugs, most commonly acetylsalicylic acid (ASA). A previous COVID-19 infection was diagnosed in 1.9% of all the patients (range 0–3.3%) (Table 2).

When examining the time-dependent distribution between vaccinations and RVOD, we observed no accumulated events within the first 4 weeks after SARS-CoV-2 vaccination (Appendix A and Figure 2), regardless of the disease or vaccine administered (Appendix A). Within the last 4 weeks before RVOD, a vaccination was recorded in 49 (50%) of the CRVO patients, 27 (48%) of the BRVO patients, 14 (33%) CRAO patients, 27 (51%) of the BRAO patients, and 38 (54%) of the AION patients. No clear relationship between vaccination and RVOD can be deduced from this data.

Data from 321 RVOD patients with a previous vaccination are presented. Eighty-nine patients were not vaccinated and are thus not described in this figure. Data from 11 patients with an incomplete vaccination time point are not included. 

### 3.2. Case–Control Analysis with Data from the Gutenberg Health Study

The COVID-19 vaccination status was similar between subjects with RVOD and age- and sex-matched population-based controls in our overall analysis, as well as for the RVOD subgroups (*n* = 327 in each arm, as there was no match for all the RVOD cases; Table 3 and Appendix A). In the arm with occlusive events, 93 (28%) patients had a CRVO, 63 (19%) a BRVO, 39 (12%) a CRAO, 46 (14%) a BRAO, and 85 (25%) suffered from AION. One hundred and thirty-six (41.6%) of the patients and one hundred and twenty-five (38.2%) of the control group were vaccinated within the previous four weeks (Table 2). 

We compared the probability of being vaccinated in the previous 4 weeks between RVOD patients and the population-based GHS sample (Table 3). The case–control study integrating population-based data from the GHS yielded no evidence of an increased risk after COVID-19 vaccination within the last 4 weeks (OR = 0.93; 95% CI: 0.60–1.45, *p* = 0.75) (Table 3). Further adjustment for the diseases with the most complete data on diabetes, obesity, arterial hypertension, smoking, and the use of anticoagulation did not alter this finding (Figure 3). A sensitivity analysis of the cases with symptom onset <2 weeks resulted in an OR = 1.05 (95% CI: 0.74–1.50; *p* = 0.79; *n* = 492) in the unadjusted analysis and OR = 0.83 (95% CI: 0.51–1.35; *p* = 0.45; *n* = 386) in the adjusted analysis. There was no significant temporal shift forward when comparing the vaccination time point between the cases and controls (spearman rho = −0.07, *p* = 0.11).

## 4. Discussion

In the present study, we surveyed RVOD patients and ascertained their COVID-19 vaccination status in a representative cross-sectional sample from different regions in Germany. Within the first 4 weeks after COVID-19 vaccination, there was no evidence for a relationship between COVID-19 vaccination and RVOD onset when we compared our data with the Gutenberg Health Study’s (GHS) results. 

At the time of data collection, four different vaccines were being administered in Germany (BNT162b2 (BioNTech/Pfizer), ChadOx1 (AstraZeneca), mRNA-1273 (Moderna), and Ad26.COV2.S (Johnson & Johnson)), and the vaccination rates were at a high level according to the Robert Koch Institute (RKI) [14]. In detail, an average of 690,106 ± 337,885 vaccinations were applied daily in June and July 2021 (range 124,965–1,432,636), both first (mean 255,168 ± 163,601; range 38,487–624,287) and second vaccinations (mean 434,866 ± 194,178; range 86,464–880,986). The RKI data suggest that, in Germany, 87% were vaccinated at least once and 49% vaccinated twice at that time [14]. A third vaccination was not recommended in Germany during the study period.

Ophthalmologic side effects after vaccination against SARS-CoV-2 have been reported sporadically, mostly in case reports or small case series, and compromise Bell´s palsy, inflammation in the anterior and posterior segments, corneal transplant rejection, retinal vascular changes, and others. Studying the retina is ideal for detecting microvascular anomalies in patients with COVID-19 disease [17,18,19] or after vaccination [2], since diagnostic tools nowadays enable multimodal high-resolution imaging of the retina and the retinal and choroidal vasculature. 

We included all patients who reported a new onset of ophthalmologic RVOD symptoms and were diagnosed with one of the RVOD subtypes (CRVO, BRVO, CRAO, BRAO, and AION) in any of the participating clinics across Germany in June and July 2021. Each of RVOD´s entities triggers typical clinical signs, can be diagnosed accurately, and the symptoms are immediately noticed and reported by patients. The high number of patients in our study who presented to an ophthalmologist within the first two weeks after the initial symptoms (76.9%) support this observation.

The time point of the onset of vascular side effects after the COVID-19 vaccination varies in the literature or is often unknown. We decided to use the four-week period as suggested by Schulz and co-workers [1]. Eighty-eight percent of all our patients presented at one of the participating study centres within four weeks after experiencing the first ocular symptoms (about 50% of all our RVOD patients were vaccinated within a four-week period before RVOD onset). However, it cannot be ruled out that late-onset side effects may also occur after vaccination. While several case series suggested a causal mechanism between RVOD and COVID-19 vaccination [3,4,20,21,22], the mechanisms behind it are not understood and highly speculative. Our data, however, failed to indicate any relationship between COVID-19 vaccination and RVOD incidence.

In our *case–control study*, we compared the patients with RVOD to healthy controls from the general population recruited by the GHS. The proportion of COVID-19 vaccinated subjects in the last 4 weeks were similar between both groups in the unadjusted analysis (Table 2 and Appendix A). In the unadjusted conditional logistic regression analysis, however, we noted one significant association, indicating a lower risk for CRAO after vaccination (Figure 3 and Table 3); nevertheless, after adjustment for the cardiovascular risk factors, there was no significant association. Thus, this finding should be carefully discussed and might be better explained by the CRAO patients’ smaller sample size and the higher subjects’ ages, along with a different cardiovascular risk profile in this subgroup, rather than a real protective effect from the vaccine. Moreover, most CRAO patients were vaccinated more than 6 weeks before RVOD symptom onset, which makes a direct effect of the vaccination on CRAO onset even more unlikely.

Our patient group’s ages and gender distributions corresponded closely to previously published data on patients with retinal vascular occlusion [23,24,25]. Regarding the ocular risk factors, 10% of our RVOD patients also had glaucoma, in line with previous studies [26,27,28]. In addition, serious cardiovascular risk factors that raise the likelihood of retinal occlusive diseases were also frequently observed in our study population [29,30,31,32,33,34].

Interestingly, the rate of arterial hypertension in our cohort was twice as high (54.7–85.5% in the RVOD subgroups) compared to the overall German population’s (31.8%) [35]. Furthermore, atrial fibrillation was three times higher in our cohort than the study subjects from the GHS (3.1% vs. 11.5%) [36]. Additionally, the rate of diabetic patients was higher in our RVOD cohort (10.8–30% in the subgroups vs. 8.9%) [37]. 

The other risk factors were comparable to the risk factors in the overall German population: proportion of smokers in our group (12.6%; range 9.4–23.2%) vs. 26.2% [38,39] and carotid artery stenosis in our cohort 18.5% vs. 6–15% (German population >65 years) [40]. In addition, 39.7% of all our RVOD patients were on anticoagulation therapy. 

Overall, our patients’ risk profile is comparable to the cardiovascular risk profiles in RVOD patients we reported recently [30,31].

The limitations of our study are the uncontrollable data quality from the individual centres and various group sizes for the vaccines applied. In addition, the pandemic with public restrictions could have led to fewer patient visits, and the actual incidences of RVOD could be higher. The data from Germany showed a 34% reduction in the diagnosis of retinal artery occlusion and AION during the pandemic, whereas the USA data revealed no change [41,42]. Furthermore, serious, life-threating, or fatal vaccine-related adverse events such as myopericarditis, deep cerebral venous thrombosis, or death [43] might have more accurately registered than ophthalmologic events. 

We further limited our analysis on a questionnaire and did not gather morphological or functional data on these patients. Future studies can also implement new technologies such as optical coherence tomography angiography, which enables the contactless and rapid examination of retinal vessel flow and vessel density. Furthermore, there is the potential risk of COVID-19 vaccination after RVOD onset, leading to a bias. To minimize this risk, we performed a sensitivity analysis including only those cases presenting within 2 weeks after symptom onset, which showed similar findings to the overall analysis. The controls were sampled from a regional population-based study (Mainz and the surrounding area), while the cases were collected all over Germany. This may have an effect on our estimations; nevertheless, the cardiovascular risk profile of the GHS is comparable to other German surveys [44].

The strengths of our study included a large sample size and data acquisition from many specialised eye clinics across Germany which limits the risk of regional clustering. In addition, by running different analyses (study cohort and controls from a population-based study), we compensated for the naturally occurring variations in the data. Especially, the case–control study shows no association.

Possible side effects of new therapies that impact a patient´s safety must be seriously monitored and reported. Especially when compared with the age- and sex-matched GHS data, we found no evidence of a causal relationship. Finally, in our multicentre study on the RVOD onset and COVID-19 vaccination status, we found no increased risk of retinal vascular occlusion.

## Figures and Tables

**Figure 1 jcm-11-05101-f001:**
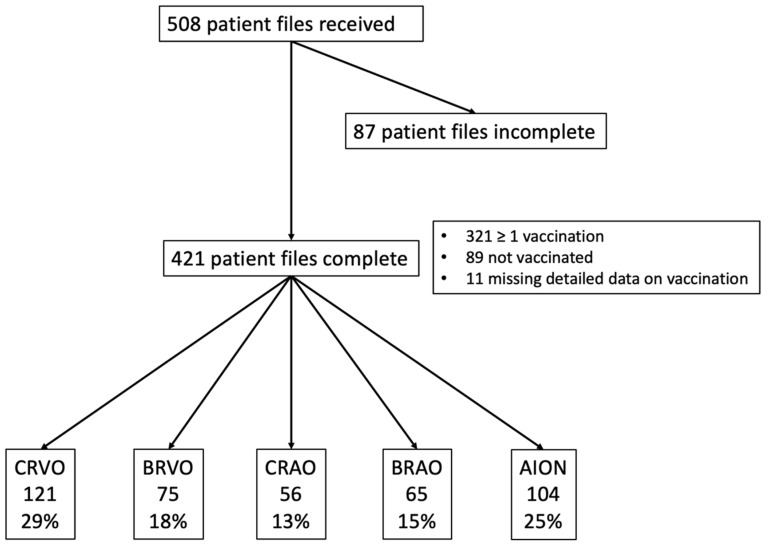
Flow chart of the included patients. CRVO = Central Retinal Vein Occlusion; BRVO = Branch Retinal Vein Occlusion; CRAO = Central Retinal Artery Occlusion; BRAO = Branch Retinal Artery Occlusion; AION = Anterior Ischaemic Optic Neuropathy.

**Figure 2 jcm-11-05101-f002:**
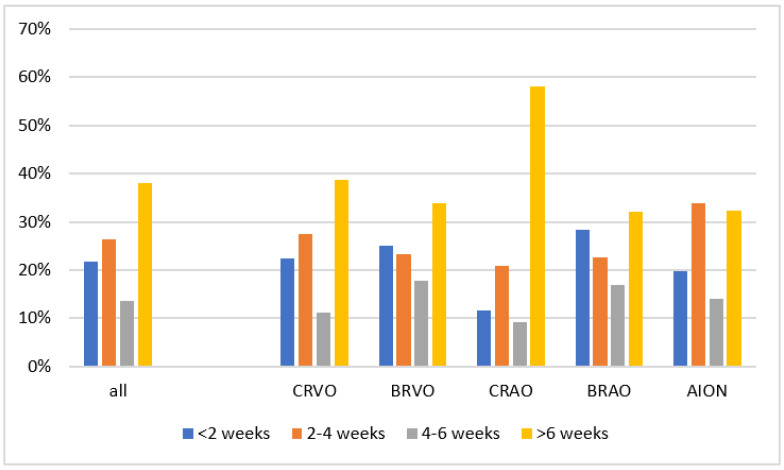
Frequency distribution of the time points of symptom onset after vaccination in the RVOD subgroups. CRVO = Central Retinal Vein Occlusion; BRVO = Branch Retinal Vein Occlusion; CRAO = Central Retinal Artery Occlusion; BRAO = Branch Retinal Artery Occlusion; AION = Anterior Ischaemic Optic Neuropathy.

**Figure 3 jcm-11-05101-f003:**
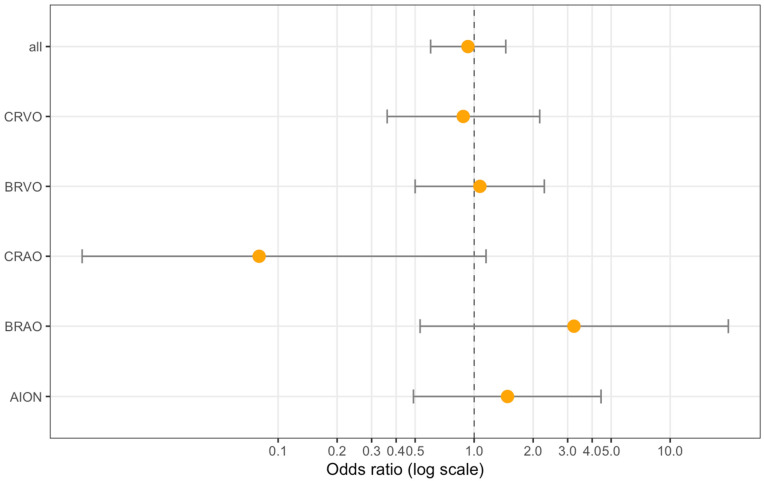
Forest plot of the association analysis between the SARS-CoV-2 vaccination status in subjects with RVOD (4 weeks prior to diagnosis) compared to population-based age- and sex-matched controls from the Gutenberg Health Study. A conditional logistic regression analysis with adjustment for diabetes, obesity, arterial hypertension, smoking, and the use of anticoagulation. CRVO = Central Retinal Vein Occlusion; BRVO = Branch Retinal Vein Occlusion; CRAO = Central Retinal Artery Occlusion; BRAO = Branch Retinal Artery Occlusion; AION = Anterior Ischaemic Optic Neuropathy.

**Table 1 jcm-11-05101-t001:** Characteristics of the study participants with RVOD.

	All	CRVO	BRVO	CRAO	BRAO	AION
*n*	421	121	75	56	65	104
Age (years)	67.6 ± 14.6	65.6 ± 16.0	65.4 ± 13.5	74.5 ± 9.9	67.3 ± 17.0	68.0 ± 13.2
Sex (female) (%)	51.8	49.6	41.3	58.9	55.4	55.8
Eye (OS) (%)	49.0	51.2	49.3	39.3	60.0	44.7
**Time point of presentation at eye clinics after onset of symptoms (% of patients)**						
<2 weeks	76.7	74.8	62.9	96.4	88.5	70.4
2–4 weeks	11.0	11.3	14.3	-	4.9	18.4
4–6 weeks	2.5	1.7	7.1	3.6	1.6	2.0
>6 weeks	9.8	12.2	15.7	-	4.9	9.2
**Ophthalmological data**						
Visual acuity diseased eye (LogMAR; median and interquartile range)	0.40 (0.10–1.30)	0.50 (0.20–1.00)	0.35 (0.10–0.70)	2.27 (1.15–2.27)	0.20 (0.10–0.80)	0.35 (0.10–1.23)
Visual acuity fellow eye (LogMAR)	0.10 (0.00–0.30)	0.10 (0.00–0.40)	0.10 (0.00–0.20)	0.10 (0.00–0.20)	0.10 (0.00–0.20)	0.20 (0.00–0.38)
Glaucoma (yes) (%)	10.7	15.0	12.5	7.4	7.8	7.9
**Systemic risk factors**						
Arterial hypertension (yes) (%)	64.5	58.3	60.0	85.5	54.7	69.9
Diabetes (yes) (%)	18.0	10.8	13.7	20.4	15.6	30.0
Obesity (yes) (%)	14.3	8.5	9.6	18.5	11.1	24.2
Smoking (yes) (%)	12.3	9.4	12.2	23.2	17.5	6.8
Carotid artery stenosis (yes) (%)	18.4	10.1	6.9	29.6	20.9	27.0
Atrial fibrillation (yes) (%)	11.7	14.2	8.2	14.8	12.7	9.1
Anticoagulation (yes) (%)						
All	39.7	32.2	23.5	61.1	50.0	41.8
Direct oral anticoagulants (DOAC)	11.4	9.3	5.9	16.7	15.6	12.2
Vitamin-K-dependent drugs	2.2	3.4	1.5	3.7	0	2.0
Acetylsalicylic acid (ASA)	22.6	18.6	13.2	35.2	28.1	23.5
combination	3.5	0.8	2.9	5.6	6.2	4.1
Prior COVID-19 infection (%)	1.9	0.9	0	2.1	3.2	3.3

CRVO = Central Retinal Vein Occlusion; BRVO = Branch Retinal Vein Occlusion; CRAO = Central Retinal Artery Occlusion; BRAO = Branch Retinal Artery Occlusion; AION = Anterior Ischaemic Optic Neuropathy.

**Table 2 jcm-11-05101-t002:** Vaccination status within 4 weeks before RVOD symptoms in a 2 × 2 contingency table.

		RVOD Cases	Controls
Vaccination within the last 4 weeks	No	191	202
	YES	136	125
	**overall**	**327**	**327**

**Table 3 jcm-11-05101-t003:** Association analysis between the COVID-19 vaccination status in subjects with retinal vascular occlusion (4 weeks prior to diagnosis) compared to population-based age- and sex-matched controls from the Gutenberg Health Study. A conditional regression analysis was computed in an (I) unadjusted way and (II) adjusted for diabetes, obesity, arterial hypertension, smoking, and the use of anticoagulation.

	Crude				Adjusted			
	*n*	OR	95% CI	*p*-Value	*n*	OR	95% CI	*p*-Value
**all**	654	1.15	0.84–1.58	0.38	506	0.93	0.60–1.45	0.75
**CRVO**	186	1.53	0.86–2.72	0.15	141	0.88	0.36–2.16	0.78
**BRVO**	126	1.06	0.55–2.05	0.87	106	1.07	0.50–2.28	0.98
**CRAO**	78	0.21	0.06–0.75	0.02	65	0.08	0.01–1.15	0.06
**BRAO**	94	1.86	0.74–4.66	0.19	72	3.23	0.53–19.8	0.21
**AION**	170	1.31	0.68–2.52	0.41	127	1.48	0.49–4.44	0.48

CRVO = Central Retinal Vein Occlusion; BRVO = Branch Retinal Vein Occlusion; CRAO = Central Retinal Artery Occlusion; BRAO = Branch Retinal Artery Occlusion; AION = Anterior Ischaemic Optic Neuropathy. A multivariable model for BRVO, BRAO, and AION was conducted without the use of anticoagulation due to model instability.

## Data Availability

The data presented in this study are available on request from the corresponding author. The data are not publicly available due to privacy.

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
