# Peer review of "Retinal Vascular Occlusion after COVID-19 Vaccination: More Coincidence than Causal Relationship? Data from a Retrospective Multicentre Study"

_jcm, 2022, doi:10.3390/jcm11175101_

Round 1
Reviewer 1 Report (New Reviewer)
Summary:
The aim of this case-control study was to determine if there is an association between SARS-CoV-2 vaccination and incident retinal vascular occlusive disease (RVOD). Cases were prospectively identified from 37 major clinics which offer retinal services in Germany recruited using a survey sent to 50 German Retina Society member clinics. The study conclusions were that there are no significant associations between COVID-19 vaccination in the 4 weeks preceding an RVOD event.
The paper is well presented, and the research question is important given the need to determine if Covid-19 vaccination is associated with sight-threatening retinal vascular occlusive diseases, with positive or negative findings having wide-ranging implications. Within the limitations of an observational study, the presented data are useful in guiding larger-scale prospective studies and would provide a valuable contribution to future meta-analyses which are likely required to answer the question more definitively. However, a number of amendments and clarifications are recommended to improve the manuscript and elevate its contributions.
Major amendments:
1. A checklist of items required in the reporting of the case-control study should be complete (STROBE statement) and included in the supplementary and referenced within the methods
2. Were there attempts to maximise the return of surveys/participation from the invited clinics as the attrition rate was 26% (13 /50). If so, could these be detailed, and if available, reasons, why some clinics did not or were not unable to participate.
a. Could the list of centres be amended to include those who were invited and those who participated?
3. Were controls from GHS recruited over the same 2-month temporal period as the cases?
4. There is a high exclusion rate due to incomplete data (~18%) which could materially change the analysis. More detailed reasons for why files are incomplete are required, preferably a table (or added to fig 1 ) summarising the non-missing data for participants who were excluded.
5. Was there a standardised protocol used for the diagnosis of each of the RVOD (e.g. FAF / OCT / Examination), if so, could additional details be added or otherwise any differences in diagnostic protocols between centres.
6. 21.8% of participants with an RVOD had been vaccinated only <2 weeks prior. Given the lag in the presentation of some of the vascular diseases (>2 weeks), it is possible that the actual vascular event could precede the vaccination. Could the authors comment on this possibility, and its effects on their association estimates and consider including a sensitivity analysis of those presenting <2 weeks and vaccinated >2-4 weeks.
7. Cases in this study are drawn from multiple centres but controls (age-, sex-matched) are drawn from a single centre which conducts the Gutenberg Health Study. There are some notable differences between cases and controls, specifically in regard to diabetes rates (18% vs 11%) which is a major risk factor for RVOD. Could the authors comment on the potential reasons for the observed differences and possible effects on the results?
8. Unfortunately I am still unclear as to how cases included in the case-control analysis are arrived at. Of all the RVOD cases (421), 321 have complete vaccination data. Yet 327 cases are included in the analysis, could the authors please clarify this and amend Figure 1 flowchart to show the selection steps?
9. Is there a reason why age and smoking status (+/- anticoagulant use status) were not adjusted for given their contribution (or protection) towards RVOD risk?
The authors comment that “vaccination suggests a protective effect against CRAO”. This seems rather surprising given the short duration with which this effect seems to be achieved. It is also of concern the size of the CI is rather wide suggesting this may be a type I, ‘false positive’ error. As the author's commented, CRAO patients were older than other RVOD groups and the sample size was small. I also note they had more comorbidities, and risk factors (including smoking) but anticoagulant use. Perhaps these results would differ if models were adjusted as per 9?
Minor Amendments:
1. The abstract result “The case-control analysis integrating population-based data from the GHS yielded no evidence of an increased risk after COVID-19 vaccination (OR=1.08; 95%-CI: 0.75-1.56, p=0.67” needs to add detail that this was with reference to vaccination within a 4-week window.
2. Could the authors clarify if “332 study participants (78.9%) had been vaccinated at least once before the RVOD onset” means these are the first vaccination figures?
3. It is detailed that “421 subjects were included where relevant …. And their COVID-19 vaccination status (yes/no) were available”. Later under figure 2 it details “11 patients with incomplete vaccination history not included”. What specifically was missing (time of vaccination, vaccine type?), as otherwise, the two sentences seem paradoxical.
4. If appropriate, authors could consider performing a statistical test of the trend of the frequency distributions presented as a function of vaccination time
5. In the discussion, rather than a range of vaccination/day, the proportion of the population with first and send vaccinations may be a more useful metric.
Author Response
Dear Reviewers,
Thank you for the comments and the time you spent commenting on our study.
We’ve answered all comments in a table and have marked the changes in the text.
Thank you again for all the helpful comments. We completely checked the data and made some changes in our analysis to improve the message we hope to convey.
We appreciate all the time you have spent to improve our paper.
Yours sincerely,
Nicolas Feltgen for the authors
Reviewer 1
|
Reviewers comment |
Answer |
Changes in the manuscript |
|||||||||||||||||||||||||||||||||||||||||||||||||||
|
A checklist of items required in the reporting of the case-control study should be complete (STROBE statement) and included in the supplementary and referenced within the methods |
The strobe protocol has been added for supplemental material, the literature was added (von Elm et al., PLOS medicine, October 2007 | Volume 4 | Issue 10 | e296) |
Literature was added in the text. |
|||||||||||||||||||||||||||||||||||||||||||||||||||
|
Were there attempts to maximise the return of surveys/participation from the invited clinics as the attrition rate was 26% (13 /50). If so, could these be detailed, and if available, reasons, why some clinics did not or were not unable to participate. |
The clinics were reminded twice within a 4-week time period. By failing to respond to our request to participate, we had to assume they weren’t interested, or unable to for staff shortage reasons, perhaps. At any rate, the nonparticipators never gave us any reasons for their lack of interest |
None |
|||||||||||||||||||||||||||||||||||||||||||||||||||
|
Could the list of centres be amended to include those who were invited and those who participated? |
We added those centres for the reviewer. The red centres failed to answer. But for data protection reasons, we are forbidden to publish the centres’ names. |
The centres in red did not send data:
|
|||||||||||||||||||||||||||||||||||||||||||||||||||
|
Were controls from GHS recruited over the same 2-month temporal period as the cases? |
We took the data from the controls immediately after the recruitment period of the RVOD-cases to create an age- and gender-matched control group. In the GHS, the actual time points of all vaccinations were surveyed. With this data, the vaccination in the control group was adjusted to match the timepoint of the individual RVOD-diagnosis. I.e. case 1 had the event of RVOD at June 15th, thus the control subject (age- and sex-matched) for case 1 was analysed for the time period before June 15th and all vaccinations after this date were not considered in this control subject. Otherwise, a sufficient sample for age- and sex-matched controls would not have been possible. |
We included the following description into the manuscript:
“Case-control study: in this analysis, we compared the odds of being vaccinated in the last four weeks among patients with RVOD (cases) to controls from the general population recruited by the Gutenberg Health Study (GHS) (age +/-5 years - and sex-matched). The recruitment of the controls took place between August 2021 and November 2021 (N=939). For each control, the vaccination status within the 4 weeks prior to the date of the RVOD diagnosis of the corresponding case was analysed and compared not to be affected by the shift in recruitment time.” |
|||||||||||||||||||||||||||||||||||||||||||||||||||
|
There is a high exclusion rate due to incomplete data (~18%) which could materially change the analysis. More detailed reasons for why files are incomplete are required, preferably a table (or added to fig 1) summarising the non-missing data for participants who were excluded. |
We received data from 508 patients. Of those, the data on 87 were insufficient (e.g. missing diagnosis, vaccination status). We ultimately analysed data from 421 patients. This info is presented in figure 1. The reasons for exclusion were missing data, i.e., gender, age, vaccination status, risk factors. The individual parameters are quite heterogeneous, so wo decided not to create another table. But we’ve added a sentence to this effect in the text. To report the reviewer, mean age of excluded subjects was 69.9±12.1 years compared to 67.6±14.6 years of include subjects. 58% of excluded subjects were female, while 52% of included subjects. Similar, side of diseased eye was 50% left side in excluded subjects and 49% in included subjects. |
We added in the results section: Comparing excluded and included patients, there was no significant difference regarding age, gender, time of occlusion, systemic risk factors and vaccination status. |
|||||||||||||||||||||||||||||||||||||||||||||||||||
|
Was there a standardised protocol used for the diagnosis of each of the RVOD (e.g. FAF / OCT / Examination), if so, could additional details be added or otherwise any differences in diagnostic protocols between centres. |
There was no standardised protocol. The clinics just sent their diagnoses. In Germany, we have published guidelines for the diagnosis and therapy of retinal vein occlusion, retinal artery occlusion, and anterior ischaemic optic neuropathy. [1–3] |
None |
|||||||||||||||||||||||||||||||||||||||||||||||||||
|
21.8% of participants with an RVOD had been vaccinated only <2 weeks prior. Given the lag in the presentation of some of the vascular diseases (>2 weeks), it is possible that the actual vascular event could precede the vaccination. Could the authors comment on this possibility, and its effects on their association estimates and consider including a sensitivity analysis of those presenting <2 weeks and vaccinated >2-4 weeks. |
Good point, we thank for this comment. We further analysed how many patients had a presentation time of >2 weeks after symptoms and a shorter vaccination time compared to the presentation time: this resulted in n=17 patients). We thus performed a sensitivity analysis with only those patients presenting within <2 weeks after symptoms onset. This resulted in a OR= 1.05 (95%CI: 0.73 – 1.51; p=0.78; n=480) in the unadjusted analysis and OR= 0.97 (95%-CI: 0.64 – 1.47; p=0.89; n=428) in the multivariable analysis. We have now added this in the main text.
We did not incorporate the named criteria >2-4 weeks of vaccination in the sensitivity analysis as well, as otherwise this leads to a largely reduced number of cases and controls, as the criteria vaccination >2-4 weeks has to be considered in the control group as well, but discussed this in the discussion section of the manuscript as well. |
We added in the text. Methods: Included were all patients with (I) newly diagnosed RVOD (subgroups: central retinal vein occlusion (CRVO), branch retinal vein occlusion (BRVO), central retinal artery occlusion (CRAO), branch retinal artery occlusion (BRAO), anterior ischaemic optic neuropathy (AION)), and (II) available data on COVID-19 vaccination.
Results: Sensitivity analysis of cases with symptoms onset <2 weeks resulted in a OR= 1.05 (95%CI: 0.74 – 1.50; p=0.79; n=492) in the unadjusted analysis and OR= 0.83 (95%-CI: 0.51 – 1.35; p=0.45; n=386) in the adjusted analysis. There was no significant temporal shift forward when comparing the vaccination time point between cases and controls (spearman rho= -0.07, p=0.11).
Conclusions: Furthermore, there is the potential risk of COVID-19 vaccination after RVOD-onset leading to bias. To minimize this risk, we performed a sensitivity analysis including only those cases presenting within 2 weeks after symptoms onset, which showed similar findings to the overall analysis. The controls were sampled from a regional population-based study (Mainz and surrounding area), while the cases were collected all over Germany. This may have effect on our estimated nevertheless the cardiovascular risk profile of the GHS is comparable to other German surveys. |
|||||||||||||||||||||||||||||||||||||||||||||||||||
|
Cases in this study are drawn from multiple centres but controls (age-, sex-matched) are drawn from a single centre which conducts the Gutenberg Health Study. There are some notable differences between cases and controls, specifically in regard to diabetes rates (18% vs 11%) which is a major risk factor for RVOD. Could the authors comment on the potential reasons for the observed differences and possible effects on the results? |
This is a good argument. We have now described in the discussion that the systemic risk factors are higher in the study population than in the GHS. In the case population there are only RVOD patients, while the sampled controls from the GHS are from a “regional” general population, with the matching criteria age and gender. Therefore, it is not surprising that the GHS patients present fewer systemic risk factors, as it is known that the risk profile of RVOD patients is higher corresponding to the data from the literature and representing a typical RVOD-group. Nevertheless, the GHS controls are not a sample from the general population of Germany, but a regional population (Mainz and surrounding areas). Overall, this regional population is not substantially different to other regions in Germany, thus we assume that this effect on the results is small. We have added this to the discussion section. |
We added in the text: Conclusions: The controls were sampled from a regional population-based study (Mainz and surrounding area), while the cases were collected all over Germany. This may have effect on our estimated nevertheless the cardiovascular risk profile of the GHS is comparable to other German surveys. |
|||||||||||||||||||||||||||||||||||||||||||||||||||
|
Unfortunately, I am still unclear as to how cases included in the case-control analysis are arrived at. Of all the RVOD cases (421), 321 have complete vaccination data. Yet 327 cases are included in the analysis, could the authors please clarify this and amend Figure 1 flowchart to show the selection steps? |
We thank for this comment. We re-analysed all our data and apologize for this insufficient description. We have data from 508 patients, 87 of those had incomplete records, thus they had to be excluded. From the 421 subjects, 321 patients were vaccinated, 89 patients had no COVID-19 vaccination at all and n=11 had missing data on time point of vaccination. For the 321+89 = 410 subjects, controls in the GHS were sampled, and for 327 subjects an age- and sex-matched control could be found. We have now further specified this in the text. Figure 1 has been changed to read:
|
We added in the text: 321 study participants (76.2%) had been vaccinated at least once before the RVOD onset, thereof for 100 study participants only one vaccination was documented. Most patients received BNT162b (BioNTech/Pfizer) (n=221), followed by ChadOx1 (AstraZeneca) (n=57), mRNA-1273 (Moderna) (n=21) and Ad26.COV2.S (Johnson & Johnson) (n=11; unknown vaccine n=11). It was also added in the text of Figure 1: 321 patients were included in the gender- and age-matched GHS-analysis. |
|||||||||||||||||||||||||||||||||||||||||||||||||||
|
Is there a reason why age and smoking status (+/- anticoagulant use status) were not adjusted for given their contribution (or protection) towards RVOD risk? |
Good point. Up to now, the data of the GHS on current smoking status and anticoagulant use was not available due to outstanding quality assurance, thus we were not able to include this in the model. Now, these variables underwent further quality control steps and could be integrated as adjustment variables. This is incorporated in Table 3 and Figure 3.
|
The text reads as follows:
“Conditional logistic regression analysis was computed in (I) unadjusted way and (II) adjusted for obesity (BMI >=30), diabetes, arterial hypertension, smoking, and use of anticoagulation.”
Please see Table 3 and Figure 3. |
|||||||||||||||||||||||||||||||||||||||||||||||||||
|
The authors comment that “vaccination suggests a protective effect against CRAO”. This seems rather surprising given the short duration with which this effect seems to be achieved. It is also of concern the size of the CI is rather wide suggesting this may be a type I, ‘false positive’ error. As the author's commented, CRAO patients were older than other RVOD groups and the sample size was small. I also note they had more comorbidities, and risk factors (including smoking) but anticoagulant use. Perhaps these results would differ if models were adjusted as per 9? |
We thank the reviewer for this valuable comment and fully agree that the effect against CRAE has a high risk of type I error. After further adjustment with smoking and anticoagulation use, the association between vaccination status and CRAO was no longer significant. We thus have rewritten these paragraphs and further discussed possible concerns.
In addition, we have now performed a sensitivity analysis excluding those cases with smoking and anticoagulant use resulting in similar results except for CRAO, showing no significant association anymore (unadjusted OR= 0.14; 95%CI: 0.02 – 1.16, p=0.07, adjusted OR=0.17, 05%CI: 0.01 - 2.60, p=0.20)
|
The text reads as follows:
“For each control, the vaccination status within the 4 weeks prior to the date of the RVOD diagnosis of the corresponding case was analysed and compared not to be affected by the shift in recruitment time. Conditional logistic regression analysis was computed in (I) unadjusted way and (II) adjusted for obesity (BMI >=30), diabetes, arterial hypertension, smoking, and use of anticoagulation. All RVOD cases were analysed, as were the different entities of retinal vascular occlusions separately.”
“We compared the probability of being vaccinated in the last 4 weeks between RVOD patients and population-based GHS sample (Table 3). The case-control study integrating population-based data from the GHS yielded no evidence of an increased risk after COVID-19 vaccination within the last 4 weeks (OR=0.93; 95%-CI: 0.60-1.45, p=0.75) (Table 3). Further adjustment for the diseases with the most complete data on diabetes, obesity, arterial hypertension, smoking, and use of anticoagulation did not alter this finding (Figure 3).”
“In our case-control study, we compared patients with RVOD to healthy controls from the general population recruited by the GHS. The proportion of COVID-19 vaccinated subjects in the last 4 weeks were similar between both groups in the unadjusted analysis (Table 2, Supplemental Table 2). In the unadjusted conditional logistic regression analysis, however, we noted one significant association, indicating a lower risk for CRAO after vaccination (Figure 3, Table 3), nevertheless after adjustment for cardiovascular risk factors there was no significant association. Thus, this finding should be carefully discussed and might be better explained by the CRAO patients’ smaller sample size and the higher subject´s age along with a different cardiovascular risk profile in this subgroup rather than a real protective effect from the vaccine. Moreover, most CRAO patients had been vaccinated more than 6 weeks before RVOD symptoms onset which makes a direct effect of vaccination on CRAO onset even more unlikely.” |
|||||||||||||||||||||||||||||||||||||||||||||||||||
|
The abstract result “The case-control analysis integrating population-based data from the GHS yielded no evidence of an increased risk after COVID-19 vaccination (OR=1.08; 95%-CI: 0.75-1.56, p=0.67” needs to add detail that this was with reference to vaccination within a 4-week window. |
This is true, the comment has been added in the text. |
That text was changed: “The case-control analysis integrating population-based data from the GHS yielded no evidence of an increased risk after COVID-19 vaccination (OR=0.93; 95%-CI: 0.60-1.45, p=0.75) with reference to vaccination within a 4-week window.” |
|||||||||||||||||||||||||||||||||||||||||||||||||||
|
Could the authors clarify if “321 study participants (78.9%) had been vaccinated at least once before the RVOD onset” means these are the first vaccination figures? |
We thank for this comment and have now added, the number of first and second vaccination before the RVOD onset. There were 100 patients with only one vaccination before RVOD onset reported, while the other patients had two vaccinations been reported.
|
The text reads as follows:
“321 study participants (76.2%) had been vaccinated at least once before the RVOD onset, thereof for 100 study participants only one vaccination was documented.” |
|||||||||||||||||||||||||||||||||||||||||||||||||||
|
It is detailed that “421 subjects were included where relevant …. And their COVID-19 vaccination status (yes/no) were available”. Later under figure 2 it details “11 patients with incomplete vaccination history not included”. What specifically was missing (time of vaccination, vaccine type?), as otherwise, the two sentences seem paradoxical.
|
The missing data concerned information about the type of vaccine and the time point of vaccination. Patients could be included as long as all the other parameters were complete and there was a statement about COVID-19 vaccination: yes/no. We made some changes in the manuscript to clarify this. |
In the results section we wrote: 321 study participants (76.2%) had been vaccinated at least once before the RVOD onset. Most patients received BNT162b (BioNTech/Pfizer) (n=221), followed by ChadOx1 (AstraZeneca) (n=57), mRNA-1273 (Moderna) (n=21) and Ad26.COV2.S (Johnson & Johnson) (n=11; unknown vaccine n=11; not vaccinated n=89). |
|||||||||||||||||||||||||||||||||||||||||||||||||||
|
If appropriate, authors could consider performing a statistical test of the trend of the frequency distributions presented as a function of vaccination time |
This is a good point. But it is not enough to assess the temporal accumulation of events in this case, as COVID-19 vaccinations were conducted in a dynamic way within this given time frame, which cannot be captured in real numbers. Therefore, in the case-control study, we compared a possible temporal shift in vaccinations in our case group compared with the control group. As results, there was no temporal trend forward, i.e., to an early occlusion event after vaccination (Spearman rho= -0.07, p=0.11). The frequency distribution is:
|
We added in the text: Sensitivity analysis of cases with symptoms onset <2 weeks resulted in an OR= 1.05 (95%CI: 0.74 – 1.50; p=0.79; n=492) in the unadjusted analysis and OR= 0.83 (95%-CI: 0.51 – 1.35; p=0.45; n=386) in the adjusted analysis. There was no significant temporal shift forward when comparing the vaccination time point between cases and controls (spearman rho= -0.07, p=0.11). |
|||||||||||||||||||||||||||||||||||||||||||||||||||
|
In the discussion, rather than a range of vaccination/day, the proportion of the population with first and second vaccinations may be a more useful metric. |
We added the numbers of patients who had got their first and second vaccinations in Germany at that time (obtained from the Robert-Koch-Institute). |
We added in the text: 87% had at least 1 vaccination and 49% were vaccinated twice at that time. |
|||||||||||||||||||||||||||||||||||||||||||||||||||
|
Reviewer 2 |
|
|
|||||||||||||||||||||||||||||||||||||||||||||||||||
|
Reviewers comment |
Answer |
Changes |
|||||||||||||||||||||||||||||||||||||||||||||||||||
|
First of all, it is stated that the study was designed prospectively, but case-control studies are retrospective in nature. A prospective case-control study is not possible. |
We thank for this comment and agree that this is a retrospective study. We made the necessary changes in the text and abstract. |
Done |
|||||||||||||||||||||||||||||||||||||||||||||||||||
|
In the statistics section, combining descriptive case-by-case analysis with a case-control study is a novel method for increasing the statistical power of the study. Aside from a well-designed case-control study, descriptive statistics have little meaning in an article whose main topic is to solve the coincidental or causality problem. |
We thank for this critical comment and discussion. We agree, that the descriptive case-by-case analysis is less meaningful than the presented case-control study. Nevertheless, as there is a recent study about the time-dependent relationship between the AstraZeneca vaccination and onset of central sinus vein thrombosis [4], we aim to present this data as well. If we would have found an association between COVID-19 vaccination and RVOD, a time-dependent relationship might have given further hints on causality. Thus, our data is rather of descriptive nature, not supporting any association. |
None |
|||||||||||||||||||||||||||||||||||||||||||||||||||
|
In order to strengthen the statistical part of the study, it would be more understandable to explain the case control study in a more understandable and detailed manner by clearly explaining how many people Gutenberg and his own cohorts took, how many were vaccinated and how many were not, and presenting them with a 2x2 contingency table or as in the graphic below. |
We thank for this comment. We have now explained the case-control study in more detail. In the corresponding time frame (August to November 2021) after the recruitment of the RVOD cases. 939 subjects were examined in the Gutenberg Health Study in this time frame. With an age range of +/- 5 years and sex-matching, pairs of 327 subjects between RVOD cases and controls could be found.
This information and the corresponding vaccination status in the 2x2 contingency table are now presented in the manuscript.
|
The following text was added:
„Case-control study: in this analysis, we compared the odds of being vaccinated in the last four weeks among patients with RVOD (cases) to controls from the general population recruited by the Gutenberg Health Study (GHS) (age +/-5 years - and sex-matched). The recruitment of the controls took place between August 2021 and November 2021 (N=939). For each control, the vaccination status within the 4 weeks prior to the date of the RVOD diagnosis of the corresponding case was analysed and compared not to be affected by the shift in recruitment time.“ |
|||||||||||||||||||||||||||||||||||||||||||||||||||
|
On the other hand, there are no exclusion criteria in the study. It may be more meaningful to conduct this study without including some cases with risk factors for RVOD. In logistic regression analysis, although there are many cardiovascular risk factors that cause RVOD, only covariant analysis of obesity diabetes and arterial hypertension needs explanation. |
The only exclusion criterion was an age below 18 years. Since the study was retrospective and we were looking for any association, we excluded as little as possible. We focused in the covariant analysis on obesity, diabetes and arterial hypertension because we had the most data on those diseases. After further quality assurance steps in the Gutenberg Health Study, also data on smoking and anticoagulation use is available Thus, we have integrated this in the analysis and also added this in the text.
In addition, we performed a sensitivity analysis excluding those subjects with relevant risk factors (smoking, anticoagulant use), resulting in similar findings. This analysis further showed that CRAO was not significantly associated (unadjusted OR= 0.14; 95%CI: 0.02 – 1.16, p=0.07, adjusted OR=0.17, 05%CI: 0.01 - 2.60, p=0.20) |
We added in the results: “For each control, the vaccination status within the 4 weeks prior to the date of the RVOD diagnosis of the corresponding case was analysed and compared not to be affected by the shift in recruitment time. Conditional logistic regression analysis was computed in (I) unadjusted way and (II) adjusted for obesity (BMI >=30), diabetes, arterial hypertension, smoking, and use of anticoagulation. All RVOD cases were analysed, as were the different entities of retinal vascular occlusions separately.”
“We compared the probability of being vaccinated in the last 4 weeks between RVOD patients and population-based GHS sample (Table 3). The case-control study integrating population-based data from the GHS yielded no evidence of an increased risk after COVID-19 vaccination within the last 4 weeks (OR=0.93; 95%-CI: 0.60-1.45, p=0.75) (Table 3). Further adjustment for the diseases with the most complete data on diabetes, obesity, arterial hypertension, smoking, and use of anticoagulation did not alter this finding (Figure 3).”
“In our case-control study, we compared patients with RVOD to healthy controls from the general population recruited by the GHS. The proportion of COVID-19 vaccinated subjects in the last 4 weeks were similar between both groups in the unadjusted analysis (Table 2, Supplemental Table 2). In the unadjusted conditional logistic regression analysis, however, we noted one significant association, indicating a lower risk for CRAO after vaccination (Figure 3, Table 3), nevertheless after adjustment for cardiovascular risk factors there was no significant association. Thus, this finding should be carefully discussed and might be better explained by the CRAO patients’ smaller sample size and the higher subject´s age along with a different cardiovascular risk profile in this subgroup rather than a real protective effect from the vaccine. Moreover, most CRAO patients had been vaccinated more than 6 weeks before RVOD symptoms onset which makes a direct effect of vaccination on CRAO onset even more unlikely.” |
|||||||||||||||||||||||||||||||||||||||||||||||||||
|
In the study, COVID infection and COVID vaccination were discussed together. It is known that infection causes thromboembolic events and this situation is accepted by the society. However, the question of whether a human-made vaccine causes thromboembolic events is ethically very important. Therefore, it would be a clearer article not to mention the infection- related parts in order to highlight the effects of vaccines in the study. While reading the article, the transition from vaccine to disease, from disease to vaccine distracts the mind and pushes the main question in the background. |
Good comment. We’ve removed the parts about COVID-19 disease. |
Done |
|||||||||||||||||||||||||||||||||||||||||||||||||||
|
English language |
We again asked a native speaker and very experienced proof-reader to make some corrections. |
Done |
|||||||||||||||||||||||||||||||||||||||||||||||||||
|
expert biostatistician |
Dr. Irene Schmidtmann is a professional biostatistician from the Institute of Medical Biostatistics, Epidemiology and Informatics of the University Medical Center Mainz. |
Done |
Literature
- van Oterendorp, C.; Lagrèze, W.A.; Feltgen, N. [Non-arteritic Anterior Ischaemic Optic Neuropathy: Pathogenesis and Therapeutic Approaches]. Klin Monbl Augenheilkd 2019, 236, 1283–1291, doi:10.1055/a-0972-1625.
- Feltgen, N.; Pielen, A. Retinaler Venenverschluss: Epidemiologie, Einteilung und klinische Befunde. Ophthalmologe 2015, 112, 607–620, doi:10.1007/s00347-015-0105-8.
- Feltgen, N.; Pielen, A. Retinaler Arterienverschluss. Ophthalmologe 2017, 114, 177–190, doi:10.1007/s00347-016-0432-4.
- Schulz, J.B.; Berlit, P.; Diener, H.; Gerloff, C.; Greinacher, A.; Klein, C.; Petzold, G.C.; Piccininni, M.; Poli, S.; Röhrig, R.; et al. COVID‐19 Vaccine‐Associated Cerebral Venous Thrombosis in Germany. Ann Neurol 2021, 10.1002/ana.26172, doi:10.1002/ana.26172.

Reviewer 2 Report (New Reviewer)

Author Response
Dear Reviewers,
Thank you for the comments and the time you spent commenting on our study.
We’ve answered all comments in a table and have marked the changes in the text.
Thank you again for all the helpful comments. We completely checked the data and made some changes in our analysis to improve the message we hope to convey.
We appreciate all the time you have spent to improve our paper.
Yours sincerely,
Nicolas Feltgen for the authors
Reviewer 1
|
Reviewers comment |
Answer |
Changes in the manuscript |
|||||||||||||||||||||||||||||||||||||||||||||||||||
|
A checklist of items required in the reporting of the case-control study should be complete (STROBE statement) and included in the supplementary and referenced within the methods |
The strobe protocol has been added for supplemental material, the literature was added (von Elm et al., PLOS medicine, October 2007 | Volume 4 | Issue 10 | e296) |
Literature was added in the text. |
|||||||||||||||||||||||||||||||||||||||||||||||||||
|
Were there attempts to maximise the return of surveys/participation from the invited clinics as the attrition rate was 26% (13 /50). If so, could these be detailed, and if available, reasons, why some clinics did not or were not unable to participate. |
The clinics were reminded twice within a 4-week time period. By failing to respond to our request to participate, we had to assume they weren’t interested, or unable to for staff shortage reasons, perhaps. At any rate, the nonparticipators never gave us any reasons for their lack of interest |
None |
|||||||||||||||||||||||||||||||||||||||||||||||||||
|
Could the list of centres be amended to include those who were invited and those who participated? |
We added those centres for the reviewer. The red centres failed to answer. But for data protection reasons, we are forbidden to publish the centres’ names. |
The centres in red did not send data:
|
|||||||||||||||||||||||||||||||||||||||||||||||||||
|
Were controls from GHS recruited over the same 2-month temporal period as the cases? |
We took the data from the controls immediately after the recruitment period of the RVOD-cases to create an age- and gender-matched control group. In the GHS, the actual time points of all vaccinations were surveyed. With this data, the vaccination in the control group was adjusted to match the timepoint of the individual RVOD-diagnosis. I.e. case 1 had the event of RVOD at June 15th, thus the control subject (age- and sex-matched) for case 1 was analysed for the time period before June 15th and all vaccinations after this date were not considered in this control subject. Otherwise, a sufficient sample for age- and sex-matched controls would not have been possible. |
We included the following description into the manuscript:
“Case-control study: in this analysis, we compared the odds of being vaccinated in the last four weeks among patients with RVOD (cases) to controls from the general population recruited by the Gutenberg Health Study (GHS) (age +/-5 years - and sex-matched). The recruitment of the controls took place between August 2021 and November 2021 (N=939). For each control, the vaccination status within the 4 weeks prior to the date of the RVOD diagnosis of the corresponding case was analysed and compared not to be affected by the shift in recruitment time.” |
|||||||||||||||||||||||||||||||||||||||||||||||||||
|
There is a high exclusion rate due to incomplete data (~18%) which could materially change the analysis. More detailed reasons for why files are incomplete are required, preferably a table (or added to fig 1) summarising the non-missing data for participants who were excluded. |
We received data from 508 patients. Of those, the data on 87 were insufficient (e.g. missing diagnosis, vaccination status). We ultimately analysed data from 421 patients. This info is presented in figure 1. The reasons for exclusion were missing data, i.e., gender, age, vaccination status, risk factors. The individual parameters are quite heterogeneous, so wo decided not to create another table. But we’ve added a sentence to this effect in the text. To report the reviewer, mean age of excluded subjects was 69.9±12.1 years compared to 67.6±14.6 years of include subjects. 58% of excluded subjects were female, while 52% of included subjects. Similar, side of diseased eye was 50% left side in excluded subjects and 49% in included subjects. |
We added in the results section: Comparing excluded and included patients, there was no significant difference regarding age, gender, time of occlusion, systemic risk factors and vaccination status. |
|||||||||||||||||||||||||||||||||||||||||||||||||||
|
Was there a standardised protocol used for the diagnosis of each of the RVOD (e.g. FAF / OCT / Examination), if so, could additional details be added or otherwise any differences in diagnostic protocols between centres. |
There was no standardised protocol. The clinics just sent their diagnoses. In Germany, we have published guidelines for the diagnosis and therapy of retinal vein occlusion, retinal artery occlusion, and anterior ischaemic optic neuropathy. [1–3] |
None |
|||||||||||||||||||||||||||||||||||||||||||||||||||
|
21.8% of participants with an RVOD had been vaccinated only <2 weeks prior. Given the lag in the presentation of some of the vascular diseases (>2 weeks), it is possible that the actual vascular event could precede the vaccination. Could the authors comment on this possibility, and its effects on their association estimates and consider including a sensitivity analysis of those presenting <2 weeks and vaccinated >2-4 weeks. |
Good point, we thank for this comment. We further analysed how many patients had a presentation time of >2 weeks after symptoms and a shorter vaccination time compared to the presentation time: this resulted in n=17 patients). We thus performed a sensitivity analysis with only those patients presenting within <2 weeks after symptoms onset. This resulted in a OR= 1.05 (95%CI: 0.73 – 1.51; p=0.78; n=480) in the unadjusted analysis and OR= 0.97 (95%-CI: 0.64 – 1.47; p=0.89; n=428) in the multivariable analysis. We have now added this in the main text.
We did not incorporate the named criteria >2-4 weeks of vaccination in the sensitivity analysis as well, as otherwise this leads to a largely reduced number of cases and controls, as the criteria vaccination >2-4 weeks has to be considered in the control group as well, but discussed this in the discussion section of the manuscript as well. |
We added in the text. Methods: Included were all patients with (I) newly diagnosed RVOD (subgroups: central retinal vein occlusion (CRVO), branch retinal vein occlusion (BRVO), central retinal artery occlusion (CRAO), branch retinal artery occlusion (BRAO), anterior ischaemic optic neuropathy (AION)), and (II) available data on COVID-19 vaccination.
Results: Sensitivity analysis of cases with symptoms onset <2 weeks resulted in a OR= 1.05 (95%CI: 0.74 – 1.50; p=0.79; n=492) in the unadjusted analysis and OR= 0.83 (95%-CI: 0.51 – 1.35; p=0.45; n=386) in the adjusted analysis. There was no significant temporal shift forward when comparing the vaccination time point between cases and controls (spearman rho= -0.07, p=0.11).
Conclusions: Furthermore, there is the potential risk of COVID-19 vaccination after RVOD-onset leading to bias. To minimize this risk, we performed a sensitivity analysis including only those cases presenting within 2 weeks after symptoms onset, which showed similar findings to the overall analysis. The controls were sampled from a regional population-based study (Mainz and surrounding area), while the cases were collected all over Germany. This may have effect on our estimated nevertheless the cardiovascular risk profile of the GHS is comparable to other German surveys. |
|||||||||||||||||||||||||||||||||||||||||||||||||||
|
Cases in this study are drawn from multiple centres but controls (age-, sex-matched) are drawn from a single centre which conducts the Gutenberg Health Study. There are some notable differences between cases and controls, specifically in regard to diabetes rates (18% vs 11%) which is a major risk factor for RVOD. Could the authors comment on the potential reasons for the observed differences and possible effects on the results? |
This is a good argument. We have now described in the discussion that the systemic risk factors are higher in the study population than in the GHS. In the case population there are only RVOD patients, while the sampled controls from the GHS are from a “regional” general population, with the matching criteria age and gender. Therefore, it is not surprising that the GHS patients present fewer systemic risk factors, as it is known that the risk profile of RVOD patients is higher corresponding to the data from the literature and representing a typical RVOD-group. Nevertheless, the GHS controls are not a sample from the general population of Germany, but a regional population (Mainz and surrounding areas). Overall, this regional population is not substantially different to other regions in Germany, thus we assume that this effect on the results is small. We have added this to the discussion section. |
We added in the text: Conclusions: The controls were sampled from a regional population-based study (Mainz and surrounding area), while the cases were collected all over Germany. This may have effect on our estimated nevertheless the cardiovascular risk profile of the GHS is comparable to other German surveys. |
|||||||||||||||||||||||||||||||||||||||||||||||||||
|
Unfortunately, I am still unclear as to how cases included in the case-control analysis are arrived at. Of all the RVOD cases (421), 321 have complete vaccination data. Yet 327 cases are included in the analysis, could the authors please clarify this and amend Figure 1 flowchart to show the selection steps? |
We thank for this comment. We re-analysed all our data and apologize for this insufficient description. We have data from 508 patients, 87 of those had incomplete records, thus they had to be excluded. From the 421 subjects, 321 patients were vaccinated, 89 patients had no COVID-19 vaccination at all and n=11 had missing data on time point of vaccination. For the 321+89 = 410 subjects, controls in the GHS were sampled, and for 327 subjects an age- and sex-matched control could be found. We have now further specified this in the text. Figure 1 has been changed to read:
|
We added in the text: 321 study participants (76.2%) had been vaccinated at least once before the RVOD onset, thereof for 100 study participants only one vaccination was documented. Most patients received BNT162b (BioNTech/Pfizer) (n=221), followed by ChadOx1 (AstraZeneca) (n=57), mRNA-1273 (Moderna) (n=21) and Ad26.COV2.S (Johnson & Johnson) (n=11; unknown vaccine n=11). It was also added in the text of Figure 1: 321 patients were included in the gender- and age-matched GHS-analysis. |
|||||||||||||||||||||||||||||||||||||||||||||||||||
|
Is there a reason why age and smoking status (+/- anticoagulant use status) were not adjusted for given their contribution (or protection) towards RVOD risk? |
Good point. Up to now, the data of the GHS on current smoking status and anticoagulant use was not available due to outstanding quality assurance, thus we were not able to include this in the model. Now, these variables underwent further quality control steps and could be integrated as adjustment variables. This is incorporated in Table 3 and Figure 3.
|
The text reads as follows:
“Conditional logistic regression analysis was computed in (I) unadjusted way and (II) adjusted for obesity (BMI >=30), diabetes, arterial hypertension, smoking, and use of anticoagulation.”
Please see Table 3 and Figure 3. |
|||||||||||||||||||||||||||||||||||||||||||||||||||
|
The authors comment that “vaccination suggests a protective effect against CRAO”. This seems rather surprising given the short duration with which this effect seems to be achieved. It is also of concern the size of the CI is rather wide suggesting this may be a type I, ‘false positive’ error. As the author's commented, CRAO patients were older than other RVOD groups and the sample size was small. I also note they had more comorbidities, and risk factors (including smoking) but anticoagulant use. Perhaps these results would differ if models were adjusted as per 9? |
We thank the reviewer for this valuable comment and fully agree that the effect against CRAE has a high risk of type I error. After further adjustment with smoking and anticoagulation use, the association between vaccination status and CRAO was no longer significant. We thus have rewritten these paragraphs and further discussed possible concerns.
In addition, we have now performed a sensitivity analysis excluding those cases with smoking and anticoagulant use resulting in similar results except for CRAO, showing no significant association anymore (unadjusted OR= 0.14; 95%CI: 0.02 – 1.16, p=0.07, adjusted OR=0.17, 05%CI: 0.01 - 2.60, p=0.20)
|
The text reads as follows:
“For each control, the vaccination status within the 4 weeks prior to the date of the RVOD diagnosis of the corresponding case was analysed and compared not to be affected by the shift in recruitment time. Conditional logistic regression analysis was computed in (I) unadjusted way and (II) adjusted for obesity (BMI >=30), diabetes, arterial hypertension, smoking, and use of anticoagulation. All RVOD cases were analysed, as were the different entities of retinal vascular occlusions separately.”
“We compared the probability of being vaccinated in the last 4 weeks between RVOD patients and population-based GHS sample (Table 3). The case-control study integrating population-based data from the GHS yielded no evidence of an increased risk after COVID-19 vaccination within the last 4 weeks (OR=0.93; 95%-CI: 0.60-1.45, p=0.75) (Table 3). Further adjustment for the diseases with the most complete data on diabetes, obesity, arterial hypertension, smoking, and use of anticoagulation did not alter this finding (Figure 3).”
“In our case-control study, we compared patients with RVOD to healthy controls from the general population recruited by the GHS. The proportion of COVID-19 vaccinated subjects in the last 4 weeks were similar between both groups in the unadjusted analysis (Table 2, Supplemental Table 2). In the unadjusted conditional logistic regression analysis, however, we noted one significant association, indicating a lower risk for CRAO after vaccination (Figure 3, Table 3), nevertheless after adjustment for cardiovascular risk factors there was no significant association. Thus, this finding should be carefully discussed and might be better explained by the CRAO patients’ smaller sample size and the higher subject´s age along with a different cardiovascular risk profile in this subgroup rather than a real protective effect from the vaccine. Moreover, most CRAO patients had been vaccinated more than 6 weeks before RVOD symptoms onset which makes a direct effect of vaccination on CRAO onset even more unlikely.” |
|||||||||||||||||||||||||||||||||||||||||||||||||||
|
The abstract result “The case-control analysis integrating population-based data from the GHS yielded no evidence of an increased risk after COVID-19 vaccination (OR=1.08; 95%-CI: 0.75-1.56, p=0.67” needs to add detail that this was with reference to vaccination within a 4-week window. |
This is true, the comment has been added in the text. |
That text was changed: “The case-control analysis integrating population-based data from the GHS yielded no evidence of an increased risk after COVID-19 vaccination (OR=0.93; 95%-CI: 0.60-1.45, p=0.75) with reference to vaccination within a 4-week window.” |
|||||||||||||||||||||||||||||||||||||||||||||||||||
|
Could the authors clarify if “321 study participants (78.9%) had been vaccinated at least once before the RVOD onset” means these are the first vaccination figures? |
We thank for this comment and have now added, the number of first and second vaccination before the RVOD onset. There were 100 patients with only one vaccination before RVOD onset reported, while the other patients had two vaccinations been reported.
|
The text reads as follows:
“321 study participants (76.2%) had been vaccinated at least once before the RVOD onset, thereof for 100 study participants only one vaccination was documented.” |
|||||||||||||||||||||||||||||||||||||||||||||||||||
|
It is detailed that “421 subjects were included where relevant …. And their COVID-19 vaccination status (yes/no) were available”. Later under figure 2 it details “11 patients with incomplete vaccination history not included”. What specifically was missing (time of vaccination, vaccine type?), as otherwise, the two sentences seem paradoxical.
|
The missing data concerned information about the type of vaccine and the time point of vaccination. Patients could be included as long as all the other parameters were complete and there was a statement about COVID-19 vaccination: yes/no. We made some changes in the manuscript to clarify this. |
In the results section we wrote: 321 study participants (76.2%) had been vaccinated at least once before the RVOD onset. Most patients received BNT162b (BioNTech/Pfizer) (n=221), followed by ChadOx1 (AstraZeneca) (n=57), mRNA-1273 (Moderna) (n=21) and Ad26.COV2.S (Johnson & Johnson) (n=11; unknown vaccine n=11; not vaccinated n=89). |
|||||||||||||||||||||||||||||||||||||||||||||||||||
|
If appropriate, authors could consider performing a statistical test of the trend of the frequency distributions presented as a function of vaccination time |
This is a good point. But it is not enough to assess the temporal accumulation of events in this case, as COVID-19 vaccinations were conducted in a dynamic way within this given time frame, which cannot be captured in real numbers. Therefore, in the case-control study, we compared a possible temporal shift in vaccinations in our case group compared with the control group. As results, there was no temporal trend forward, i.e., to an early occlusion event after vaccination (Spearman rho= -0.07, p=0.11). The frequency distribution is:
|
We added in the text: Sensitivity analysis of cases with symptoms onset <2 weeks resulted in an OR= 1.05 (95%CI: 0.74 – 1.50; p=0.79; n=492) in the unadjusted analysis and OR= 0.83 (95%-CI: 0.51 – 1.35; p=0.45; n=386) in the adjusted analysis. There was no significant temporal shift forward when comparing the vaccination time point between cases and controls (spearman rho= -0.07, p=0.11). |
|||||||||||||||||||||||||||||||||||||||||||||||||||
|
In the discussion, rather than a range of vaccination/day, the proportion of the population with first and second vaccinations may be a more useful metric. |
We added the numbers of patients who had got their first and second vaccinations in Germany at that time (obtained from the Robert-Koch-Institute). |
We added in the text: 87% had at least 1 vaccination and 49% were vaccinated twice at that time. |
|||||||||||||||||||||||||||||||||||||||||||||||||||
|
Reviewer 2 |
|
|
|||||||||||||||||||||||||||||||||||||||||||||||||||
|
Reviewers comment |
Answer |
Changes |
|||||||||||||||||||||||||||||||||||||||||||||||||||
|
First of all, it is stated that the study was designed prospectively, but case-control studies are retrospective in nature. A prospective case-control study is not possible. |
We thank for this comment and agree that this is a retrospective study. We made the necessary changes in the text and abstract. |
Done |
|||||||||||||||||||||||||||||||||||||||||||||||||||
|
In the statistics section, combining descriptive case-by-case analysis with a case-control study is a novel method for increasing the statistical power of the study. Aside from a well-designed case-control study, descriptive statistics have little meaning in an article whose main topic is to solve the coincidental or causality problem. |
We thank for this critical comment and discussion. We agree, that the descriptive case-by-case analysis is less meaningful than the presented case-control study. Nevertheless, as there is a recent study about the time-dependent relationship between the AstraZeneca vaccination and onset of central sinus vein thrombosis [4], we aim to present this data as well. If we would have found an association between COVID-19 vaccination and RVOD, a time-dependent relationship might have given further hints on causality. Thus, our data is rather of descriptive nature, not supporting any association. |
None |
|||||||||||||||||||||||||||||||||||||||||||||||||||
|
In order to strengthen the statistical part of the study, it would be more understandable to explain the case control study in a more understandable and detailed manner by clearly explaining how many people Gutenberg and his own cohorts took, how many were vaccinated and how many were not, and presenting them with a 2x2 contingency table or as in the graphic below. |
We thank for this comment. We have now explained the case-control study in more detail. In the corresponding time frame (August to November 2021) after the recruitment of the RVOD cases. 939 subjects were examined in the Gutenberg Health Study in this time frame. With an age range of +/- 5 years and sex-matching, pairs of 327 subjects between RVOD cases and controls could be found.
This information and the corresponding vaccination status in the 2x2 contingency table are now presented in the manuscript.
|
The following text was added:
„Case-control study: in this analysis, we compared the odds of being vaccinated in the last four weeks among patients with RVOD (cases) to controls from the general population recruited by the Gutenberg Health Study (GHS) (age +/-5 years - and sex-matched). The recruitment of the controls took place between August 2021 and November 2021 (N=939). For each control, the vaccination status within the 4 weeks prior to the date of the RVOD diagnosis of the corresponding case was analysed and compared not to be affected by the shift in recruitment time.“ |
|||||||||||||||||||||||||||||||||||||||||||||||||||
|
On the other hand, there are no exclusion criteria in the study. It may be more meaningful to conduct this study without including some cases with risk factors for RVOD. In logistic regression analysis, although there are many cardiovascular risk factors that cause RVOD, only covariant analysis of obesity diabetes and arterial hypertension needs explanation. |
The only exclusion criterion was an age below 18 years. Since the study was retrospective and we were looking for any association, we excluded as little as possible. We focused in the covariant analysis on obesity, diabetes and arterial hypertension because we had the most data on those diseases. After further quality assurance steps in the Gutenberg Health Study, also data on smoking and anticoagulation use is available Thus, we have integrated this in the analysis and also added this in the text.
In addition, we performed a sensitivity analysis excluding those subjects with relevant risk factors (smoking, anticoagulant use), resulting in similar findings. This analysis further showed that CRAO was not significantly associated (unadjusted OR= 0.14; 95%CI: 0.02 – 1.16, p=0.07, adjusted OR=0.17, 05%CI: 0.01 - 2.60, p=0.20) |
We added in the results: “For each control, the vaccination status within the 4 weeks prior to the date of the RVOD diagnosis of the corresponding case was analysed and compared not to be affected by the shift in recruitment time. Conditional logistic regression analysis was computed in (I) unadjusted way and (II) adjusted for obesity (BMI >=30), diabetes, arterial hypertension, smoking, and use of anticoagulation. All RVOD cases were analysed, as were the different entities of retinal vascular occlusions separately.”
“We compared the probability of being vaccinated in the last 4 weeks between RVOD patients and population-based GHS sample (Table 3). The case-control study integrating population-based data from the GHS yielded no evidence of an increased risk after COVID-19 vaccination within the last 4 weeks (OR=0.93; 95%-CI: 0.60-1.45, p=0.75) (Table 3). Further adjustment for the diseases with the most complete data on diabetes, obesity, arterial hypertension, smoking, and use of anticoagulation did not alter this finding (Figure 3).”
“In our case-control study, we compared patients with RVOD to healthy controls from the general population recruited by the GHS. The proportion of COVID-19 vaccinated subjects in the last 4 weeks were similar between both groups in the unadjusted analysis (Table 2, Supplemental Table 2). In the unadjusted conditional logistic regression analysis, however, we noted one significant association, indicating a lower risk for CRAO after vaccination (Figure 3, Table 3), nevertheless after adjustment for cardiovascular risk factors there was no significant association. Thus, this finding should be carefully discussed and might be better explained by the CRAO patients’ smaller sample size and the higher subject´s age along with a different cardiovascular risk profile in this subgroup rather than a real protective effect from the vaccine. Moreover, most CRAO patients had been vaccinated more than 6 weeks before RVOD symptoms onset which makes a direct effect of vaccination on CRAO onset even more unlikely.” |
|||||||||||||||||||||||||||||||||||||||||||||||||||
|
In the study, COVID infection and COVID vaccination were discussed together. It is known that infection causes thromboembolic events and this situation is accepted by the society. However, the question of whether a human-made vaccine causes thromboembolic events is ethically very important. Therefore, it would be a clearer article not to mention the infection- related parts in order to highlight the effects of vaccines in the study. While reading the article, the transition from vaccine to disease, from disease to vaccine distracts the mind and pushes the main question in the background. |
Good comment. We’ve removed the parts about COVID-19 disease. |
Done |
|||||||||||||||||||||||||||||||||||||||||||||||||||
|
English language |
We again asked a native speaker and very experienced proof-reader to make some corrections. |
Done |
|||||||||||||||||||||||||||||||||||||||||||||||||||
|
expert biostatistician |
Dr. Irene Schmidtmann is a professional biostatistician from the Institute of Medical Biostatistics, Epidemiology and Informatics of the University Medical Center Mainz. |
Done |
Literature
- van Oterendorp, C.; Lagrèze, W.A.; Feltgen, N. [Non-arteritic Anterior Ischaemic Optic Neuropathy: Pathogenesis and Therapeutic Approaches]. Klin Monbl Augenheilkd 2019, 236, 1283–1291, doi:10.1055/a-0972-1625.
- Feltgen, N.; Pielen, A. Retinaler Venenverschluss: Epidemiologie, Einteilung und klinische Befunde. Ophthalmologe 2015, 112, 607–620, doi:10.1007/s00347-015-0105-8.
- Feltgen, N.; Pielen, A. Retinaler Arterienverschluss. Ophthalmologe 2017, 114, 177–190, doi:10.1007/s00347-016-0432-4.
- Schulz, J.B.; Berlit, P.; Diener, H.; Gerloff, C.; Greinacher, A.; Klein, C.; Petzold, G.C.; Piccininni, M.; Poli, S.; Röhrig, R.; et al. COVID‐19 Vaccine‐Associated Cerebral Venous Thrombosis in Germany. Ann Neurol 2021, 10.1002/ana.26172, doi:10.1002/ana.26172.

Round 2
Reviewer 1 Report (New Reviewer)
Thank you to the authors who have carefully considered and amended the manuscript in light of the suggested revisions. There are improvements notable throughout the manuscript, and the authors provide comprehensive replies to all my comments. I have enjoyed reading the revised paper!
Reviewer 2 Report (New Reviewer)
All questions were well-answered by the authors.
This manuscript is a resubmission of an earlier submission. The following is a list of the peer review reports and author responses from that submission.
Round 1
Reviewer 1 Report
This article tried to assess the correlations between Covid-19 vaccination and retinal vascular occlusion.
Although the idea is potentially interesting, there are several limitations irremediably affecting the study.
First of all, this was a retrospective analysis including a limited area of patients using questionnaires and not considering the prospective evaluation. From this point of view, I strongly suggest the authors to come back to study design in order to obtain more interesting and robust data from their analyses.
The parameters the authors tested are very sensitive to arbitrary evaluation and poorly specific. I did not understand why the authors target all vascular occlusions and did not limit them to specific diseases. I think this is a very important point. The mechanism of vascular occlusion is not clear, and the study did not find any significance It is clear that the importance of this study is very low.
The description of the statistical analysis is inadequate and it requires major modifications. The exclusion criteria were not described in depth.
I strongly suggest the authors come back to study design, considering a more robust collection and analysis of data.
Reviewer 2 Report
In this paper, the authors investigate whether vaccination against SARS-CoV2 is associated with retinal vascular occlusive disease. They retrieved their study cohort of patients with retinal occlusive diseases from a survey done in Germany, Europe, and compared this population with data from the population-based Gutenberg Health Study (GHS). They found no clear relationship between vaccination and RVOD.
I congratulate the authors for analyzing the data for a large cohort study in Germany and for tackling this important health issue.
However, I spot on major concerns:
· A survey is not an appropriate statistical method to investigate associations. Cross sectional comprehensive studies are needed to verify associations between odds of an event and exposure to risk factors. Try to design a 2x2 contingency table and then start from there. Apparently, the group Exposed but not diseased is missing.
· The time frame of the survey is very short, only two months.
· Exclusion criteria are not provided.
· In both the introduction and the discussion, the author mixed COVID infection and COVID vaccination. This is not correct, as different pathophysiology has been hypothesized; please revise.
· Abstract: what do the author mean with specialized clinic?
· “Retinal findings covered uveitis, central serous chorioretinopathy, and acute retinal necrosis.” A reference is missing.
· “is an increased probability of RVOD onset shortly after the vaccination versus onset with delay after the vaccination” This sentence is not clear. Please provide explanation on what you mean with early after vaccination and delay after vaccination.
· How did you select adjusting for confounding factors?